# Beyond Treatment Decisions: The Predictive Value of Comprehensive Geriatric Assessment in Older Cancer Patients

**DOI:** 10.3390/cancers17152489

**Published:** 2025-07-28

**Authors:** Eleonora Bergo, Marina De Rui, Chiara Ceolin, Pamela Iannizzi, Chiara Curreri, Maria Devita, Camilla Ruffini, Benedetta Chiusole, Alessandra Feltrin, Giuseppe Sergi, Antonella Brunello

**Affiliations:** 1Hospital Psychology, Veneto Institute of Oncology IOV—IRCCS, 35128 Padua, Italy; eleonora.bergo@iov.veneto.it (E.B.); alessandra.feltrin@iov.veneto.it (A.F.); 2Division of Geriatrics, DIDAS Medicine Department, University Hospital of Padua, 35128 Padua, Italy; marina.derui@aopd.veneto.it (M.D.R.); chiara.ceolin.1@studenti.unipd.it (C.C.); chiara.curreri@aopd.veneto.it (C.C.); maria.devita@unipd.it (M.D.); giuseppe.sergi@unipd.it (G.S.); 3Department of Medicine—DIMED, University of Padua, 35128 Padua, Italy; 4Aging Research Center, Department of Neurobiology, Care Sciences and Society, Karolinska Institutet and Stockholm University, 17165 Solna, Stockholm, Sweden; 5Department of General Psychology—DPG, University of Padua, 35131 Padua, Italy; 6Department of Oncology, Medical Oncology 1, Veneto Institute of Oncology IOV—IRCCS, 35128 Padua, Italy; camilla.ruffini@iov.veneto.it (C.R.); benedetta.chiusole@iov.veneto.it (B.C.); antonella.brunello@iov.veneto.it (A.B.)

**Keywords:** cancer, geriatric depression scale, mini-mental state examination, geriatric, neuropsychology, older patients

## Abstract

The increasing life expectancy poses a number of challenges for clinicians, particularly in oncology. Evaluating frailty among older patients is essential for predicting the outcomes of oncological treatments. This study aimed to identify the elements of the Comprehensive Geriatric Assessment (CGA) that most influence decisions about anti-cancer treatment for older patients and explore the CGA’s predictive value for mortality. We identified the key CGA elements that aid clinicians in making therapeutic decisions regarding anti-cancer treatments for older adults, as well as their role as independent predictors of mortality. Functional capacity was identified as a significant predictor of mortality within four years, alongside geriatric depression and living arrangements. These findings emphasize the importance of multidimensional geriatric assessments in the management of older cancer patients.

## 1. Introduction

The aging population presents several challenges for clinicians, particularly in oncology. With an increasing life expectancy, more older individuals diagnosed with cancer are seeking healthcare, raising complex issues related to their clinical and psychological management [1]. The complexity of the aging process makes it difficult to predict the outcomes of oncological treatments in older patients [2,3]. Compared to younger people, these patients often have distinct characteristics, including multimorbidities, polypharmacy, reduced functional reserve, and socio-environmental challenges. These factors significantly impact survival, treatment toxicity, and quality of life, complicating the risk–benefit assessment of cancer therapies. In this context, frailty emerges as a crucial consideration in assessing older patients with cancer [4]. Frailty reflects an increased vulnerability to stressors, resulting in a higher risk of adverse health outcomes. Due to the fact that frailty influences treatment tolerance, prognosis, and quality of life, its accurate assessment is essential for personalized care and appropriate therapeutic decision-making in oncology also [4].

Recognizing the importance of these factors, since 2005, the International Society of Geriatric Oncology (SIOG) has emphasized the need for a Comprehensive Geriatric Assessment (CGA) in of older cancer patients [5]. However, the systematic implementation of CGA in routine clinical practice remains challenging. In everyday practice, oncologists frequently use tools such as the Eastern Cooperative Oncology Group Performance Status (ECOG PS) [6] or the Karnofsky Performance Score (KPS) [7] to estimate patients’ general condition. For frailty screening, the Geriatrics 8 (G8) [8] is commonly employed to quickly identify old cancer patients at risk of frailty-related poor health status. Balducci’s criteria have been commonly used to classify older patients into three risk groups (fit, vulnerable, and frail) based on factors such as age, comorbidities, and functional status [9]. Its importance lies in its ability to provide a rapid yet comprehensive assessment, guiding therapeutic decisions in geriatric oncology [10]. CGA-based scores such as the Onco-MPI have also been proven to be useful in predicting short-term mortality among older cancer patients. Globally, CGA has been shown to help clinicians in assessing prognoses, predicting treatment toxicity, and guiding treatment decisions.

Indeed, recently milestone trials have provided strong evidence for the role of CGA-based interventions, particularly in reducing chemotherapy toxicity. In particular, both the GAP70+ trial [11] and the GAIN trial [2] demonstrated that vulnerable older patients who received geriatric intervention beyond chemotherapy had significantly lower serious treatment toxicity, compared to those managed by standard oncological practice. In this study, following the CGA, in the intervention group, more patients received a primary dose reduction, indicating an impact on treatment decisions, but importantly, reduced dose intensity did not compromise survival, which was similar between groups at one year [11].

However, despite robust evidence and clear guidelines emphasizing the critical role of CGA in the evaluation of older oncology patients, significant gaps persist in both research and the clinical application of CGA, and its systematic implementation in oncological clinical practice remains limited, often due to time and resource constraints [2,12,13] The current limitations also encompass the scarcity of clinical trials involving older patients [14], and the absence of large-scale prospective studies validating the effectiveness of CGA in enhancing long-term outcomes.

In light of these challenges and research gaps, this study aims to contribute to the existing literature with the following objectives: (I) to retrospectively investigate which specific elements of CGA are associated with antitumor treatment decisions for old patients, aiming to identify the most relevant factors for clinical practice, and (II) to explore the predictive role of various CGA elements on four-year mortality among a cohort of old cancer patients, with the objective of enhancing risk stratification and personalized care.

## 2. Materials and Methods

### 2.1. Study Population

This retrospective observational study included older patients consecutively recruited at the Veneto Institute of Oncology (IOV) between 2003 to 2023, with follow-up available.

The inclusion criteria were as follows: (i) patients aged 70 years or older with a newly diagnosed, histologically confirmed solid or hematological cancer; (ii) having been seen at their first oncology visit; (iii) a G8 score of 14 or less; and (iv) having a minimum follow-up period of 4 years.

The study protocol was approved by the local Ethics Committee (Ethics Committee of the IOV) and complied with the guidelines of the Declaration of Helsinki.

### 2.2. Data Collection

The following information was collected by trained physicians.

#### 2.2.1. G8 Questionnaire

As illustrated in Figure 1, and in accordance with the inclusion criteria, all older patients presenting for their first visit at the IOV were administered the G8 questionnaire by a trained psychologist. The G8 questionnaire is a validated, brief screening tool for oncology populations, known for its predictive value regarding functional decline and overall survival [15], and it is internationally recognized as a potential first-line screening instrument. The cut-off score is set at 14 out of 17 items. If a patient’s G8 score falls below the cut-off of 14, a more comprehensive assessment is mandated through the CGA conducted by the psychologist. Subsequently, the oncologist, in accordance with the geriatrician, reviewed the patient, completing the CGA with any missing clinical data.

#### 2.2.2. CGA

The CGA takes an estimated 20 min to complete and consists of the following tools:*Activities of Daily Living—ADL* for the assessment of functional capacity [16].*Instrumental Activity of Daily Life—IADL* for the assessment of instrumental capacity [17].*Geriatric Depression Scale—short form—GDS* to assess the emotional status [18].*Mental State Examination—MMSE* [19] and *Short Portable Mental Status Questionnaire—SPMSQ* [20] to assess cognitive performance.*Exton Smith scale—ESS* to determine the risk of developing pressure sores [21].*The Mini Nutritional Assessment (MNA)* to determine nutritional status [22].*The Cumulative Illness Rating Scale (CIRS)* for the assessment of comorbidities [23].

#### 2.2.3. Balducci Evaluation

At the conclusion of the final multidimensional evaluation, patients were categorized into three risk groups based on Balducci’s criteria: frail, vulnerable, and fit (Table 1) [24].

*Covariates*: Data on civil and cohabitation status, total number of medications, cancer site, tumor status, and KPS were collected. Participants’ body weight and height were measured while they wore light indoor clothing and no shoes. The body mass index (BMI) was then calculated as the ratio of weight to height squared (kg/m^2^).

### 2.3. Statistical Analysis

Categorical variables are reported as counts and percentages. Continuous quantitative variables are presented as means ± standard deviations or medians (interquartile ranges), and the Shapiro–Wilk test was used to assess their normal distribution. Patients were divided into two groups through a median split on (77 years). Variables between groups (considering both sex differences and age groups) were compared using the Mann–Whitney U test and the Kruskal–Wallis test for quantitative variables, and the chi-squared test for categorical variables. In order to identify factors influencing treatment choices a decision tree was built using the chi-squared automatic interaction detection (CHAID) algorithm [25]. Variables with a *p*-value < 0.10 in multivariate logistic regression were included in the CHAID analysis. The decision tree is a non-parametric procedure that requires no assumptions about the underlying data. We set the maximum number of splits to four, the minimum number of cases in the parent node to 50, and the minimum number of cases in the child node to 20 to preserve statistical power. Node splitting was considered significant with a *p*-value < 0.05 using Bonferroni’s correction. As the proportion of missing data for each variable was insignificant, multiple imputations were not used. The final model was evaluated by calculating the misclassification risk estimate and overall accuracy percentage. The misclassification risk for the sample was confirmed by conducting a tenfold cross-validation. The misclassification risk, which refers to the incorrect classification of a patient, is estimated by applying the tree to an excluded subsample. Survival curves were estimated using the Kaplan–Meier method, stratifying participants based on GDS values (≥10 or <10). The same analyses were conducted considering age groups. Differences between the survival curves in the two groups were compared using the log-rank test. To analyze predictors of mortality, with a focus on components of the CGA, a Cox regression analysis was performed. Multivariate analyses were performed, adjusting for age, sex, tumor site, cancer treatment, and the presence of metastasis. The analyses were also performed while stratifying by sex and age groups. All analyses were performed using the Statistical Package for Social Science (SPSS) 29.0 software (IBM Corp., Armonk, NY, USA) with the significance level set to *p* < 0.05.

## 3. Results

### 3.1. Characteristics of the Sample at the Baseline

The sample characteristics by sex differences are detailed in Table 2. From the initial 8560 patients enrolled, 7022 patients (3222 females) were included, with an average age of 78.3 ± 12.9 years. Women were on average older than men, more frequently widowed (34.7% vs. 10.8%, *p* < 0.001), and more often lived alone (29.5% vs. 11.0%, *p* < 0.001). Men, on the other hand, mostly had localized tumors and generally showed better results in the multidimensional geriatric assessment compared to women [MNA: 24.21 ± 3.50 vs. 21.92 ± 1.29, *p* = 0.04; IADL: 6.39 ± 2.14 vs. 6.27 ± 2.16, *p* = 0.02; GDS: 4.13 ± 2.82 vs. 4.64 ± 2.87, *p* < 0.001] despite taking a larger number of medications. Individuals living alone scored better on almost all CGA items, except for the MNA, where no significant differences were observed between those living with family members and those living alone. These differences were particularly pronounced among older patients, as those living with family members exhibited lower scores, especially in ADL and IADL tests (see Appendix A). Finally, the overall geriatric assessment indicated a higher number of frail individuals among women (27.1% vs. 23.6%, *p* < 0.001).

### 3.2. CGA Parameters and Clinical Decision Making

The CHAID decision tree (Figure 2) shows the factors that influenced the clinical decision to initiate treatment, based on multidimensional geriatric evaluation items. The primary decision was guided by ADL values. Among patients with an ADL score ≤ 3, 37.3% received treatment. In contrast, 55.8% and 64.2% of patients with ADL scores of 3–5 or ≥5, respectively, received chemotherapy or hormone therapy. The next step was the assessment of cohabitation status. People living with a spouse or partner received chemotherapy or hormone therapy more frequently (70.3% for those with an ADL score of 3–5 and 73.2% for those with an ADL score ≥ 5), whereas approximately 53% and 42% of those living alone or with other family members did not receive treatment. The third step involved evaluating chronological age: only 25% of those under 84 years old did not receive treatment, compared to 55% of those over 84 years old who were not treated. The risk estimate for the decision tree was 0.330, and the standard error was 0.011. This means that the decision to treat patients or not was predicted with an accuracy of approximately 67% by this classification tree analysis. The sensitivity and specificity of the decision tree were 84.7% and 30.8% respectively. 

### 3.3. Mortality

After four years of follow-up, 1533 participants (21.9%) from the sample had died (830 males and 703 females). The Kaplan–Meier curves, stratified by participants’ GDS values (≥10 vs. <10), depicting overall survival over a four-year period, are shown in Figure 3. Notably, patients with GDS scores ≥ 10 demonstrated significant differences in survival outcomes (log-rank *p* = 0.01). This difference was particularly pronounced in patients under 77 years of age (see Appendix A).

After adjusting for age, gender, primary tumor site and cancer treatment, and the presence of metastasis, a Cox regression analysis revealed that higher GDS scores (OR 1.04, 95% CI 1.01–1.07, *p* = 0.04) and living with family members (OR 1.67, 95% CI 1.35–2.07, *p* < 0.001) were independently associated with four-year survival (Table 3). Additionally, IADL scores were identified as protective factors. These findings were consistent among male participants, with GDS remaining significant (OR 1.04, 95% CI 1.01–1.10, *p* = 0.03) (Figure 4). In contrast, for female participants, the only significant factors associated with mortality were living with family members.

When stratifying by age groups, younger patients exhibited both the MMSE and GDS as risk factors for mortality (OR 1.08, 95% CI 1.02–1.14, *p* = 0.008 and OR 1.07, 95% CI 1.01–1.13, *p* = 0.02, respectively). In older patients, higher scores on the MNA and IADL were associated with protective outcomes (OR 0.95, 95% CI 0.92–0.99, *p* = 0.007 and OR 0.92, 95% CI 0.86–0.99, *p* = 0.04) (see Appendix A).

## 4. Discussion

Our study on a large dataset of older cancer patients assessed by means of a CGA identified a possible decision tree focused on key CGA items that can influence anti-cancer treatment decisions. Functional status emerged as the primary factor in this decision pathway, followed by living arrangements. Additionally, we explored the predictive role of various CGA elements on 4-year mortality in a cohort of older cancer patients. Our analysis seemed to suggest that GDS was a significant predictor of mortality, particularly in men and younger individuals. In contrast, among women, neither GDS nor other CGA components were significantly associated with survival outcomes. Conversely, in older patients, higher MNA and IADL scores were identified as protective factors. These results are suggestive and are not of easy clinical interpretation but underline the relevance of CGA in the management of cancer care, especially for older patients. Notably, our findings highlight the critical influence of depression, rather than cognitive decline, on survival in this population.

The CGA was originally developed by geriatricians as a multidisciplinary assessment of the older patient that includes a number of important clinical aspects [5,26]. The CGA is an in-depth assessment that could be defined as a multidisciplinary assessment in which the multiple issues of older individuals are identified, described, and explained, when possible, while evaluating and developing a coordinated care plan that focuses interventions on the person’s strengths, needs, and available services [27]. In oncology, where older patients are increasingly prevalent due to the gradual decline in organ function and the higher incidence of multimorbidity associated with aging, managing these patients requires early identification of key geriatric conditions by CGA [28]. The CGA is highly prognostic, helping to identify patients at risk of frailty and disability, and of chemotherapy tolerance and toxicity [29,30,31,32], thus enabling oncologists to make more appropriate treatment decisions tailored to each patient’s specific condition [28]. The CGA-based oncological multidimensional prognostic index (onco-MPI) also classifies patients as high-, intermediate-, or low-risk based on tumor characteristics [33]; CGA and onco-MPI scores have been shown to have prognostic value in older cancer patients, and may help with decision-making and subgroup stratification in specialized trials [9,34].

Our findings show that functional status, as measured using ADLs, was the primary factor guiding the decision-making process, followed by an assessment of the patient’s living situation. Age, on the other hand, played a secondary role and was considered only at the final stage. While the unique nature of our study makes it difficult to directly compare with previous research, the importance of functional status in the health of older patients is well documented. Not only in oncology but across various clinical settings, the ability to perform ADLs is closely linked to quality of life in older patients, making the evaluation of functional status a fundamental aspect of geriatric care [35,36]. Surprisingly, no cognitive aspects, such as the MMSE score, emerged in the decision tree. Given the importance of cognitive functions, particularly executive functions and general cognition, for treatment adherence and informed consent, we would have expected these factors to play a role in the clinician’s therapeutic decisions. In fact, patients with dementia frequently receive less care compared to patients without dementia due to challenges related to obtaining informed consent, doubts about treatment adherence, and potential difficulties in reporting adverse events [37]. And this is where geriatric assessment becomes crucial. Thanks to its multidimensional view of the patient, in fact, it becomes essential in identifying patients who are likely to benefit from therapy, ensuring that individuals with frailty in one domain are not excluded, as long as this frailty is compensated for by strengths in other domains. Thus, a low MMSE score does not necessarily influence the clinical decision to offer oncological treatment if cognitive frailty is counterbalanced by strong social support.

As regards predictors of overall survival, functional status emerges as one of the most important ones, a conclusion supported by various studies. For instance, a systematic review conducted by Couderc and colleagues found that functional status was a significant predictor of survival in nearly half of the studies analyzed [38]. Additionally, it played a crucial role in treatment decisions and postoperative complications [38]. Our study revealed gender differences in mortality predictors, consistent with the existing literature that indicates that IADL scores act as significant protective factors in men but not in women. This disparity may stem from the tendency of men with higher IADL scores to engage in health-promoting behaviors, which enhance their overall well-being. In contrast, women may rely on alternative coping strategies or robust social support systems that mitigate the adverse effects of lower IADL scores, enabling them to maintain their health despite functional limitations [39].

Furthermore, the MMSE emerged as a significant predictor of mortality exclusively in younger individuals, likely because it more accurately reflects cognitive health in this population. Among older patients, however, other clinical factors—such as comorbidities, functional status, and nutritional status—appear to exert a greater influence on mortality risk. This observation aligns with previous research, such as a three-year study of 249 older cancer patients (median age: 77 years), which identified independent survival predictors including abnormal albumin levels and high malnutrition risk [40].

Lastly, living status and GDS score emerged as important predictors of mortality in men and younger individuals but not in women or older patients. Regarding living arrangements, the majority of women in our study lived alone, while men and young people frequently resided with other family members. Living with family members may seem like a paradoxical risk factor for mortality, especially considering the existing literature [41]. This result should certainly be treated with caution and requires further investigation. More detailed data on caregiver dependency, frailty, and home care support could clarify whether this result is confounded by the indication. A further investigation of the caregiver’s role would, therefore, be desirable.

However, this is likely linked to the greater complexity of these patients, who required more assistance and had significantly lower scores across almost all components of the CGA. On the other hand, several factors may account for the differences observed in the GDS. Previous studies have highlighted significant gender differences in the development of social networks and relationships throughout life, with men’s relationships tending to be less intimate than those of women [42]. Additionally, it is culturally less acceptable for men to express their emotions compared to women [42,43]. As a result, depression in men may often be underdiagnosed. Furthermore, men may be more vulnerable to the negative health impacts of depression, primarily due to differences in coping mechanisms and societal expectations [44]. On the other hand, it is possible that, among older individuals, other health factors may overshadow the impact of depression on mortality risk [45]. We believe that these results provide significant insights into the clinical and psychological care of older cancer patients, particularly regarding the crucial impact of the GDS on mortality in younger individuals. First and foremost, it is essential to urgently implement systematic depression screening for cancer patients. Given the varying effects of depression based on gender and age, it is imperative to develop increasingly targeted interventions to address these specific needs. This also underscores the importance of forming multidisciplinary oncology care teams that include mental health professionals specializing in psycho-oncology and enhancing access to psycho-oncological services by integrating them with medical cancer care. Furthermore, it is vital to reduce the stigma associated with seeking mental health support, particularly among men, through educational programs that can promote accessibility. Finally, ongoing research and evaluation are crucial to improving psychological care for cancer patients, with a focus on conducting studies that assess the impact of psychosocial interventions on mortality rates, especially in high-risk groups identified via GDS scores.

### Strengths and Limitations

This study helps bridge the gap in research regarding the use of CGA in oncology, presenting an investigation that includes a large cohort of older patients and an extended follow-up period. This robust design enables a more thorough evaluation of the utility of CGA and its potential impact, offering a significant contribution to a field in which further research is urgently needed to validate CGA’s role in optimizing care for older patients. However, some limitations of this study must be acknowledged. First, the use of the MMSE as a cognitive assessment tool presents certain drawbacks. While the MMSE is widely utilized, its limitations are known, such as reduced sensitivity in detecting early or mild cognitive impairment (i.e., Mild Cognitive Impairment), especially in patients with higher levels of education or from specific cultural contexts. The data presented in this study were collected from 2003 onwards. This enabled a large number of participants to be enrolled but also meant that a dated test such as the Mini-Mental State Examination (MMSE) had to be used, which is less sensitive than newer tools. Additionally, the MMSE may not capture the full range of cognitive decline, which could influence the accuracy of our findings related to cognitive health and its role in predicting mortality. Furthermore, although the extended follow-up is a strength, it may introduce attrition bias, as patients with more severe conditions may have been lost to follow-up. One important limitation of the decision tree model developed in this study is its relatively low specificity (30.8%), which restricts its clinical utility, particularly in settings where false positives may lead to unnecessary interventions or resource use. The model was intentionally optimized for sensitivity in order to maximize the identification of high-risk individuals, consistent with the exploratory and screening-oriented aim of the analysis. However, this trade-off came at the cost of specificity. External validation in independent cohorts will be essential to evaluate the generalizability and robustness of any refined model. Another limitation is that our dataset does not include reliable longitudinal follow-up or systematically collected, time-stamped information that would enable a robust assessment of changes over time. Finally, the generalizability of our results may be impacted by the specific characteristics of the study population, which may not fully represent the broader diversity of older patients encountered in clinical practice.

## 5. Conclusions

This study pointed out the key elements of CGA that aid clinicians in making therapeutic decisions regarding anti-cancer treatments for older patients, as well as their role as independent predictors of mortality. While a significant emphasis was placed on functional status during the treatment decision-making process, functional capacity emerged as a major predictor of four-year mortality, alongside the GDS and living arrangements. These findings highlight the necessity of a multidimensional geriatric assessment in all aspects of managing older cancer patients.

## Figures and Tables

**Figure 1 cancers-17-02489-f001:**
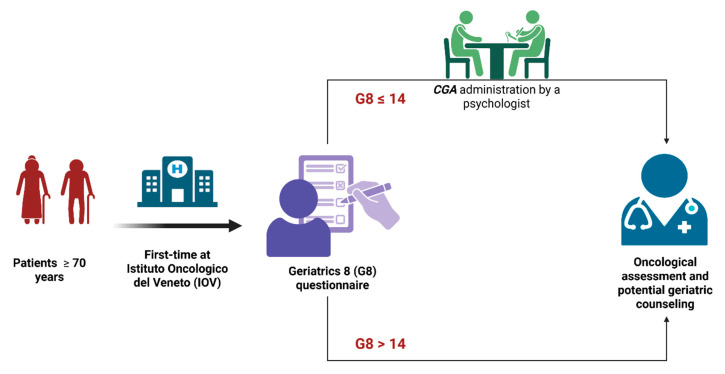
Procedure for administering the Comprehensive Geriatric Assessment (CGA) to elderly patients upon their first visit to the Istituto Oncologico Veneto (IOV). Figure created in BioRender Iannizzi P., 2025 https://BioRender.com/b72n253 (accessed on 17 July 2025).

**Figure 2 cancers-17-02489-f002:**
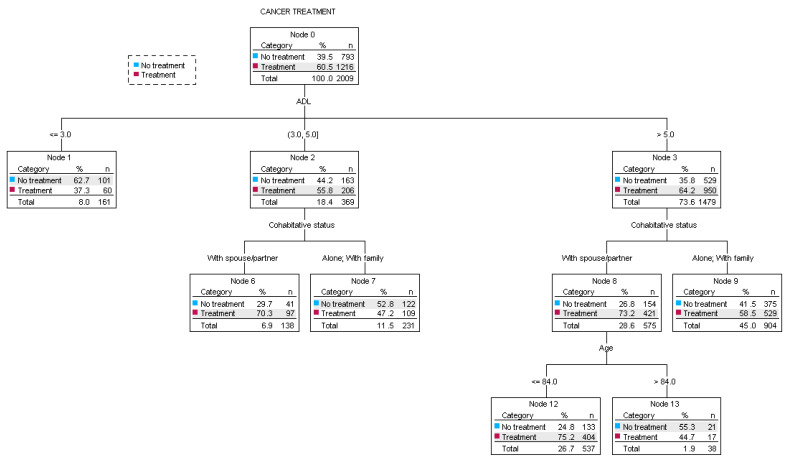
Decision tree regarding cancer treatment prescription according to geriatric multidimensional evaluation. Abbreviations: ADL: Activities of Daily Living.

**Figure 3 cancers-17-02489-f003:**
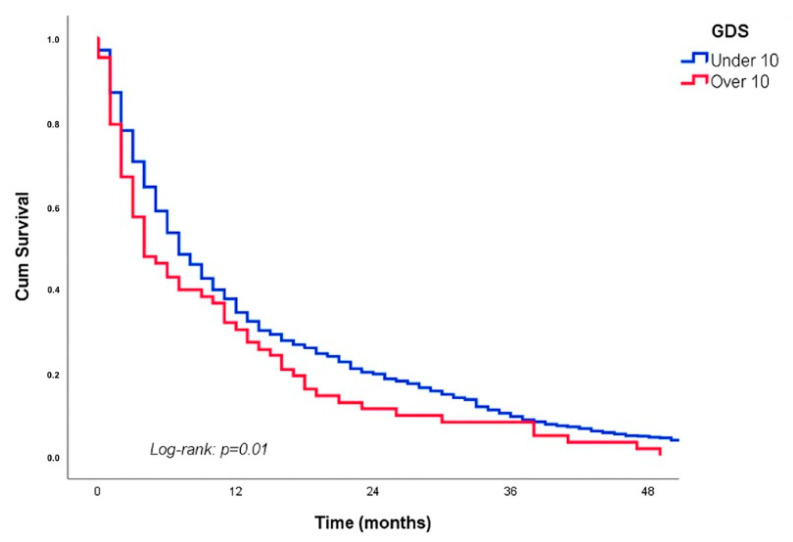
Kaplan–Meier curves of overall 4-year survival stratified by GDS scores (below 10 vs. above 10 points), presented for the entire sample. Abbreviations: GDS: Geriatric Depression Scale.

**Figure 4 cancers-17-02489-f004:**
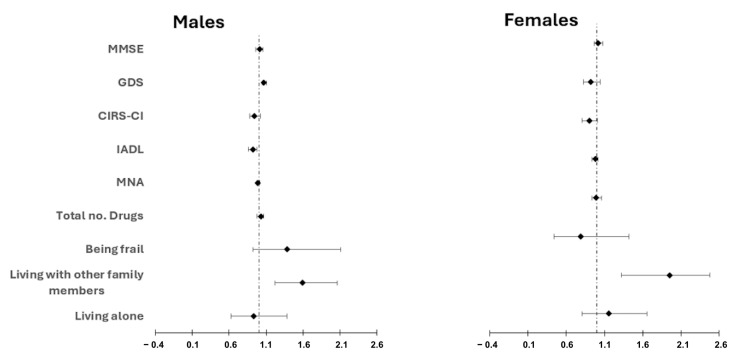
Prognostic values for overall 4-year survival of CGA elements by sex: Cox Proportional Hazards Model Forest Plot. Abbreviations: MMSE: Mini-Mental State Examination; GDS: Geriatric Depression Scale; CIRS-CI: Cumulative Illness Rating Scale-Comorbidity Index; IADL: Instrumental Activities of Daily Living; MNA: Mini Nutritional Assessment.

**Table 1 cancers-17-02489-t001:** Classification of patients according to Balducci’s criteria.

Definition	Stage
- No functional dependence in ADLs or IADLs	Fit
- No relevant comorbidities	
- No geriatric syndromes	
- Dependence in one or more IADLs but not in ADLs	Vulnerable
- Comorbidities present but not severe or life-threatening	
- May have mild depression or mild memory disorder, but not other significant geriatric syndromes	
- Age > 84 years	Frail
- Dependence in one or more ADLs	
- Any significant geriatric syndrome	
- 3 or more grade 3 comorbidities or one grade 4	

Abbreviations: ADL: Activities of Daily Living; IADL: Instrumental Activities of Daily Living.

**Table 2 cancers-17-02489-t002:** Characteristics of the sample at the baseline.

*p*-Value	Females (*n* = 3222)	Males (*n* = 3800)	Total Sample (*n* = 7022)	Variable
*0.04*	78.7 ± 13.5	78.0 ± 12.4	78.3 ± 12.9	Age
0.95	24.20 ± 6.07	24.21 ± 6.04	24.20 ± 6.06	BMI
0.38	2.45 ± 1.66	2.48 ± 1.70	2.47 ± 1.68	No. of comorbidities
*<0.001*				*Civil status*
	1117 (34.7%)	412 (10.8%)	1529 (21.8%)	Widow/widower
	1490 (46.2%)	3027 (79.7%)	4517 (64.3%)	Married
*<0.001*				*Cohabitation status*
	861 (29.5%)	399 (11.0%)	1260 (19.2%)	Alone
	381 (13.1%)	546 (15.0%)	927 (14.2%)	With partner/spouse
	1637 (56.1%)	2668 (73.5%)	4305 (65.7%)	With other family members
	40 (1.4%)	18 (0.5%)	58 (0.9%)	NH
*<0.001*				*Cancer site*
	827 (25.7%)	964 (25.4%)	1791 (25.5%)	Colorectal
	549 (17.0%)	818 (21.5%)	1367 (19.5%)	Upper digestive tract and liver
	770 (23.9%)	12 (0.3%)	782 (11.1%)	Breast
	153 (4.7%)	863 (22.7%)	1016 (14.5%)	Urinary tract
	251 (7.8%)	273 (7.2%)	524 (7.5%)	Hematological
	72 (2.2%)	105 (2.8%)	177 (2.5%)	Brain
	139 (4.3%)	179 (4.7%)	318 (4.5%)	Skin
	53 (1.6%)	123 (3.2%)	176 (2.5%)	Lung
	408 (12.7%)	463 (12.2%)	871 (12.4%)	Other *
*<0.001*				*Tumor status*
	1129 (35.0%)	1544 (40.6%)	2673 (39.9%)	Localized
	761 (23.6%)	1146 (30.2%)	1907 (28.4%)	Advanced
*<0.001*				*Treatment*
	1207 (37.4%)	920 (24.2%)	2127 (31.7%)	Surgery
	1153 (70.0%)	932 (60.3%)	2085 (65.3%)	Chemotherapy/ Endocrine/Targeted therapy
*<0.001*	674 (20.9%)	988 (26.0%)	1662 (23.8%)	*Metastasis*
*<0.001*				*Karnofsky performance scale*
	936 (29.1%)	1226 (32.3%)	2162 (30.8%)	80
	909 (28.2%)	1279 (33.7%)	2188 (31.2%)	90–100
				*Geriatric multidimensional evaluation*
*0.04*	21.92 ± 1.29	24.21 ± 3.50	23.25 ± 1.34	MNA
0.19	5.33 ± 2.01	5.36 ± 1.27	5.35 ± 1.65	ADL
*0.02*	6.27 ± 2.16	6.39 ± 2.14	6.33 ± 2.15	IADL
*<0.001*	4.64 ± 2.87	4.13 ± 2.82	4.36 ± 2.85	GDS
0.09	26.69 ± 1.77	28.15 ± 5.31	27.49 ± 4.56	MMSE
0.99	1.00 (0.0; 2.00)	1.00 (0.00; 2.00)	1.00 (0.00; 2.00)	CIRS-CI
*<0.001*	3.67 ± 2.77	3.99 ± 3.02	4.60 ± 3.26	Total drugs
*<0.001*				*Final multidimensional evaluation*
	874 (27.1%)	898 (23.6%)	1772 (25.2%)	Frail
	1077 (33.4%)	1355 (35.7%)	2432 (34.6%)	Fit
	1150 (35.7%)	1353 (35.6%)	2503 (35.6%)	Vulnerable

Notes: numbers are expressed as means (standard deviations), medians (interquartile ranges), or numbers (percentages), as appropriate. *Abbreviations*: NH: nursing homes; BMI: body mass index; MNA: Mini Nutritional Assessment; ADL: Activities of Daily Living; IADL: Instrumental Activities of Daily Living; MMSE: Mini-Mental State Examination; GDS: Geriatric Depression Scale; CIRS-CI: Cumulative Illness Rating Scale-Comorbidity Index. * Unknown primary origin (*n* = 119); gynecologic (*n* = 31); head and neck (*n* = 18); bone (*n* = 14); others (*n* = 689).

**Table 3 cancers-17-02489-t003:** Prognostic values for overall 4-year survival of CGA elements: Cox proportional-hazard models.

Adjusted Analysis				Unadjusted Analysis				Variable
*p*-Value	95% IC		HR	*p*-Value	95% IC		HR	
	Upper	Lower			Upper	Lower		
								*Cohabitative status*
0.49	1.4	0.85	1.09	0.69	4.87	0.09	0.67	Alone
*<0.001*	*2.07*	*1.35*	*1.67*	0.8	5.65	0.11	0.78	With other family members/carers
								*Multidimensional evaluation*
0.28	1.05	0.98	0.99	0.27	1.05	0.98	0.99	MNA
*<0.001*	*0.96*	*0.85*	*0.91*	*0.001*	*0.96*	*0.86*	*0.91*	IADL
0.33	1.05	0.98	1.01	0.58	1.04	0.98	1.01	MMSE
*0.04*	*1.07*	*1.01*	*1.04*	*0.04*	*1.07*	*1.01*	*1.03*	GDS
0.52	1.03	0.87	0.92	*0.03*	*1.18*	*1.09*	*1.03*	CIRS-CI
0.45	1.05	0.98	1.01	0.76	1.04	0.97	1.01	Total no. drugs
0.79	1.45	0.75	1.05	0.81	1.31	0.81	1.03	Being Frail at CGA

Abbreviations: MNA: Mini Nutritional Assessment; IADL: Instrumental Activities of Daily Living; MMSE: Mini-Mental State Examination; GDS: Geriatric Depression Scale; CIRS-CI: Cumulative Illness Rating Scale-Comorbidity Index; CGA: Comprehensive Geriatric Assessment. Analyses are adjusted for age, gender, cancer treatment and site, and metastasis.

## Data Availability

Due to privacy and ethical restrictions, the data that support the findings of this study are available on request from the corresponding author.

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
