# Peer review of "Beyond Treatment Decisions: The Predictive Value of Comprehensive Geriatric Assessment in Older Cancer Patients"

_cancers, 2025, doi:10.3390/cancers17152489_

Round 1
Reviewer 1 Report
Comments and Suggestions for Authors
Thank you for the opportunity to review this interesting and important article. While it presents valuable insights, some revisions are required prior to publication.
Introduction
- General: The term “Older adults” (or “older patients”) is used. To enhance clarity for the reader, it would be helpful to define “older adults”/ ”older patients” in the introduction.
- Page 3, lines 90-92: One can read that “However, despite robust evidence and clear guidelines emphasizing the critical role of CGA in the evaluation of older oncology patients…”. This sentence should be supported by multiple references.
Methods
- Page 4, lines 134-135: Katz ADL was used to assess functional autonomy through Activities of Daily Living (ADL). While the Katz Index is a well-established tool for evaluating basic physical self-care abilities, it does not fully capture the broader concept of functional autonomy, which also includes cognitive, social, and instrumental aspects of independent living. I therefore recommend that the authors to clarify this distinction and consider revising the wording to avoid the implication that functional autonomy in its entirety is assessed using the Katz ADL scale alone. Further, the term instrumental autonomy is used and assesses with IADL (Lawton MP, Brody EM, 1969). Does the Lawton and Brody explicit uses this term? I recommend that the authors revisit the literature to examine which terms are commonly used in relation to the instruments cited. If the current terminology does not align with the original sources or established definitions, the authors should either adjust their wording accordingly or support their usage with appropriate references.
Discussion
- Page 10, line 289: The term “functional status” is used in contrast to the method where the term functional autonomy is used. This variation in terminology may cause confusion for the reader. I therefore recommend that the authors review their use of the terms functional autonomy and functional status, and ensure consistency throughout the manuscript, supported by the referenced literature.
- I recommend that the authors to elaborate further on the clinical implications with this study.
Author Response
Thank you very much for taking the time to review this manuscript. Please find the detailed responses below and the corresponding revisions in track changes in the re-submitted files.
Comment 1: General:The term “Older adults” (or “older patients”) is used. To enhance clarity for the reader, it would be helpful to define “older adults”/ ”older patients” in the introduction.
Response 1: Thank you for highlighting this incongruence. We have decided to replace the term 'older adult' with 'older patient' so that the text is clearer.
Comment 2: Page 3, lines 90-92: One can read that “However, despite robust evidence and clear guidelines emphasizing the critical role of CGA in the evaluation of older oncology patients…”. This sentence should be supported by multiple references.
Response 2: We thank the Reviewer for his/her suggestion. We have now provided more and strong references.
Comment 3: Page 4, lines 134-135: Katz ADL was used to assess functional autonomy through Activities of Daily Living (ADL). While the Katz Index is a well-established tool for evaluating basic physical self-care abilities, it does not fully capture the broader concept of functional autonomy, which also includes cognitive, social, and instrumental aspects of independent living. I therefore recommend that the authors to clarify this distinction and consider revising the wording to avoid the implication that functional autonomy in its entirety is assessed using the Katz ADL scale alone. Further, the term instrumental autonomy is used and assesses with IADL (Lawton MP, Brody EM, 1969). Does the Lawton and Brody explicit uses this term? I recommend that the authors revisit the literature to examine which terms are commonly used in relation to the instruments cited. If the current terminology does not align with the original sources or established definitions, the authors should either adjust their wording accordingly or support their usage with appropriate references.
Comment 4: Page 10, line 289: The term “functional status” is used in contrast to the method where the term functional autonomy is used. This variation in terminology may cause confusion for the reader. I therefore recommend that the authors review their use of the terms functional autonomy and functional status, and ensure consistency throughout the manuscript, supported by the referenced literature.
Response 3-4:
Thank you for suggesting this change. Indeed, functional autonomy is determined by the various indices examined by the Comprehensive Geriatric Assessment (CGA). In clinical practice, the ADL and IADL scales have historically been used. These certainly have the limitations you have highlighted, but they provide clinicians with a strong point of reference to evaluate the autonomy. we now adopt the term functional capacity throughout the manuscript. This choice, supported by recent literature (Xu et al., 2024), more accurately conveys the construct being measured—namely, the individual’s ability to perform daily activities, without conflating this with autonomy in a broader decisional sense. For similar reasons, we now refer to instrumental capacity in place of 'instrumental autonomy'.
Reviewer 2 Report
Comments and Suggestions for Authors
Bergo and colleagues performed an observational cohort study including 7,022 older adults with newly diagnosed solid or hematological malignancies, recruited over a 20-year period. The aim was to evaluate which elements of the CGA influence oncological treatment decisions and predict four-year mortality. As such, they found that functional autonomy (ADL), cohabitation status, and age were key determinants of treatment decisions, while higher GDS scores and living with family were independently associated with increased mortality risk. The manuscript is wellwritten
Nevertheless, I have some comments:
Major:
- Low specificity of the decision tree model (30.8%) limits its clinical utility. The authors should elaborate on this limitation and discuss whether model refinement or alternative methods might yield better predictive performance.
- Cognitive assessment (MMSE/SPMSQ) did not influence treatment decisions, which is surprising given the relevance of cognition for consent and adherence. This raises concerns about the sensitivity of the tools used and deserves more critical discussion.
- The term “treatment” is used broadly. It is unclear whether treatment intent (curative vs. palliative) and modality (chemotherapy, surgery, hormonal therapy) were taken into account. More detailed stratification would strengthen the conclusions.
- The counterintuitive finding that living with family members is associated with higher mortality should be interpreted cautiously. More detailed data on caregiver dependency, frailty, or home care support could clarify whether this reflects confounding by indication.
- Some statistically significant associations (e.g., GDS and mortality with HR ~1.04) show limited clinical relevance. The authors should better balance statistical significance with clinical interpretability.
- Attrition bias and handling of missing data are insufficiently addressed. Given the long follow-up, information on loss to follow-up or censoring is essential to judge the validity of the survival analysis.
- The study spans two decades (2003–2023), yet no analyses are presented on temporal trends in CGA application or outcomes. This is a missed opportunity to explore how practice evolved over time and/or at least
- External validation of the decision tree and Cox models is lacking. Without validation in an independent cohort, the generalizability of the findings remains uncertain. (should be elaborated on)
Minor :
- The authors refer to the MMSE as the main cognitive tool but do not report full MMSE score distributions or clinically relevant thresholds.
- Some results (e.g., ADL thresholds in the decision tree) would benefit from graphical clarification or supplementary tables for transparency.
Author Response
Thank you very much for taking the time to review this manuscript. Please find the detailed responses below and the corresponding revisionsin track changes in the re-submitted files
Major
Comment 1: Low specificity of the decision tree model (30.8%) limits its clinical utility. The authors should elaborate on this limitation and discuss whether model refinement or alternative methods might yield better predictive performance.
Response 1:
We thank the Reviewer for his/her suggestion. We acknowledge the low specificity (30.8%) of the decision tree model and agree that this limits its immediate clinical applicability, particularly in contexts where false positives carry significant consequences. We have added a paragraph in the Discussion section to elaborate on this limitation. Specifically, we note that the decision tree was optimized for sensitivity to ensure high case detection at the cost of specificity, given the exploratory nature of the model. Nonetheless, we agree that refinement strategies—such as pruning complexity, adjusting class weights, or employing ensemble methods (e.g., random forests or gradient boosting)—may yield more balanced performance. Future studies will aim to explore and compare such approaches in larger and more balanced datasets.
Please see p.49 line 393: One important limitation of the decision tree model developed in this study is its relatively low specificity (30.8%), which restricts its clinical utility, particularly in settings where false positives may lead to unnecessary interventions or resource use. The model was intentionally optimized for sensitivity in order to maximize the identification of high-risk individuals, consistent with the exploratory and screening-oriented aim of the analysis. However, this trade-off came at the cost of specificity. External validation in independent cohorts will be essential to evaluate the generalizability and robustness of any refined model.
Comment 2: Cognitive assessment (MMSE/SPMSQ) did not influence treatment decisions, which is surprising given the relevance of cognition for consent and adherence. This raises concerns about the sensitivity of the tools used and deserves more critical discussion.
Response 2:
We agree that the results are inconsistent and have, in fact, tried to explain this in the discussion “Surprisingly, no cognitive aspects, such as the MMSE score, emerged in the decision tree. Given the importance of cognitive functions, particularly executive functions and general cognition, for treatment adherence and informed consent, we would have expected these factors to play a role in the clinician’s therapeutic decisions. In fact, patients with dementia frequently receive less care compared to non-demented patients due to challenges related to obtaining informed consent, doubts about treatment adherence, and potential difficulties in reporting adverse events [36 ]. And this is where geriatric assessment becomes crucial. Thanks to its multidimensional view of the patient, in fact, it becomes essential in identifying patients who are likely to benefit from therapy, ensuring that individuals with frailty in one domain are not excluded, as long as this frailty is compensated for by strengths in other domains. Thus, a low MMSE score does not necessarily influence the clinical decision to offer oncological treatment if cognitive frailty is counterbalanced by strong social support'. As suggested by the reviewer, it is possible that the result can also be attributed to the lower specificity of the instruments. The MMSE and the SPMSQ are specific to Alzheimer's dementia and are less sensitive to functions that are crucial for decision-making and therapeutic adherence, such as executive functions. Furthermore, the MMSE is less effective at diagnosing deficits in early frailty, such as those associated with mild cognitive impairment (MCI). While we acknowledge that more recent instruments (such as the Montreal Cognitive Assessment (MOCA) capture this complexity more effectively, it should be noted that the data were collected from 2003 onwards when more sophisticated instruments were unavailable, and the MMSE remains the most widely used tool in CGA.
We have now specified this in limitation more clearly in the discussion.
Please see p.49 l. 381: However, some limitations of this study must be acknowledged. First, the use of the MMSE as a cognitive assessment tool presents certain drawbacks. While the MMSE is widely utilized, it is known to have limitations, such as reduced sensitivity in detecting early or mild cognitive impairment (i.e. Mild Cognitive Impairment), especially in patients with higher levels of education or from specific cultural contexts. The data presented in this study were collected from 2003 onwards. This enabled a large number of participants to be enrolled, but also meant that a dated test such as the Mini-Mental State Examination (MMSE) had to be used, which is less sensitive than newer tools.Additionally, the MMSE may not capture the full range of cognitive decline, which could influence the accuracy of our findings related to cognitive health and its role in predicting mortality. Furthermore, although the extended follow-up is a strength, it may introduce attrition bias, as patients with more severe conditions may have been lost to follow-up.
Comment 3: The term “treatment” is used broadly. It is unclear whether treatment intent (curative vs. palliative) and modality (chemotherapy, surgery, hormonal therapy) were taken into account. More detailed stratification would strengthen the conclusions.
Response 3:
Thank you for highlighting this incongruence. The CGA assessment is carried out at our institute during the initial oncology appointment, before treatment begins. 'Treatment' refers to curative chemotherapy or hormone therapy specific to the tumour type. During treatment at the Institute, patients may undergo several lines of therapy. For this reason, we believe that stratifying by therapy type and number of lines would disperse the data too widely.
Comment 4: The counterintuitive finding that living with family members is associated with higher mortality should be interpreted cautiously. More detailed data on caregiver dependency, frailty, or home care support could clarify whether this reflects confounding by indication.
Response 4:
Thank you for highlighting this inconsistency. As mentioned in the discussion, we agree that this is an unexpected finding that should be interpreted with caution. Given the complexity of elderly people, we suggested gender differences as an explanation. The reviewer's suggestions to assess the degree of dependency of the caregiver and the quality of care in more detail are very helpful. Unfortunately, as we do not have more in-depth data, we are unable to provide further insight into the interpretation. Further studies on the caregivers of these patients, who are often overlooked, are desirable.
Comment 5: Some statistically significant associations (e.g., GDS and mortality with HR ~1.04) show limited clinical relevance. The authors should better balance statistical significance with clinical interpretability.
Response 5:
We thank the reviewer for this observation. We have tried to express more caution when interjecting these results into the discussion.
Comment 6: Attrition bias and handling of missing data are insufficiently addressed. Given the long follow-up, information on loss to follow-up or censoring is essential to judge the validity of the survival analysis.
Comment 7: The study spans two decades (2003–2023), yet no analyses are presented on temporal trends in CGA application or outcomes. This is a missed opportunity to explore how practice evolved over time and/or at least
Response 6 and 7:
We thank the reviewer for this observation. We fully agree that analyzing temporal trends over the 20-year study period would have added important insights into the evolution of CGA practices and related outcomes. Unfortunately, our dataset does not include reliable longitudinal follow-up or systematically collected time-stamped information that would allow for a robust assessment of changes over time. For this reason, we were unable to conduct temporal analyses without introducing significant bias or speculative interpretation. We now explicitly acknowledge this limitation in the Limitation section and highlight the need for future longitudinal studies that can more accurately track how CGA application and clinical practices evolve over time.
Comment 8: External validation of the decision tree and Cox models is lacking. Without validation in an independent cohort, the generalizability of the findings remains uncertain. (should be elaborated on)
Response 8:
We agree that external validation is critical for assessing the generalizability of our models. As stated in the limitations section, our analyses were based on a single-center cohort, and the decision tree and Cox models have not yet been validated on an independent dataset. We have now elaborated on this limitation (see previous response 1), emphasizing that while internal cross-validation was used to reduce overfitting, external validation in independent cohorts from other institutions or settings is essential before clinical implementation.
Minor
Comment 9: The authors refer to the MMSE as the main cognitive tool but do not report full MMSE score distributions or clinically relevant thresholds.
Comment10: The authors refer to the MMSE as the main cognitive tool but do not report full MMSE score distributions or clinically relevant thresholds.
Response 9-10
Unfortunately, the tree is an automated process and it is extremely difficult for us to modify it. However, we would be grateful if the reviewer could specify which indices they feel should be made more explicit.